# Defects in G-Actin Incorporation into Filaments in Myoblasts Derived from Dysferlinopathy Patients Are Restored by Dysferlin C2 Domains

**DOI:** 10.3390/ijms21010037

**Published:** 2019-12-19

**Authors:** Ximena Báez-Matus, Cindel Figueroa-Cares, Arlek M. Gónzalez-Jamett, Hugo Almarza-Salazar, Christian Arriagada, María Constanza Maldifassi, María José Guerra, Vincent Mouly, Anne Bigot, Pablo Caviedes, Ana M. Cárdenas

**Affiliations:** 1Centro Interdisciplinario de Neurociencia de Valparaíso, Facultad de Ciencias, Universidad de Valparaíso, Valparaíso 2360102, Chile; ximena.baez@cinv.cl (X.B.-M.); cindel.figueroa@postgrado.uv.cl (C.F.-C.); arlek.gonzjam@gmail.com (A.M.G.-J.); constanza.maldifassi@cinv.cl (M.C.M.); mjguerraf@gmail.com (M.J.G.); 2Departamento de Anatomía y Medicina Legal, Facultad de Medicina, Universidad de Chile, Santiago 8389100, Chile; 3Sorbonne Université, Inserm, Institut de Myologie, UMRS 974, Center for Research in Myology, 75013 Paris, France; vincent.mouly@upmc.fr (V.M.); anne.bigot@upmc.fr (A.B.); 4Programa de Farmacología Molecular y Clínica, ICBM, Facultad de Medicina, Universidad de Chile, Santiago 8389100, Chile; pablo.caviedes@cicef.cl; 5Centro de Biotecnología y Bioingeniería (CeBiB), Departamento de Ingeniería Química, Biotecnología y Materiales, Facultad de Ciencias Físicas y Matemáticas, Universidad de Chile, Santiago 8370456, Chile

**Keywords:** dysferlin, actin, C2 domains, annexin A2, dysferlinopathy

## Abstract

Dysferlin is a transmembrane C-2 domain-containing protein involved in vesicle trafficking and membrane remodeling in skeletal muscle cells. However, the mechanism by which dysferlin regulates these cellular processes remains unclear. Since actin dynamics is critical for vesicle trafficking and membrane remodeling, we studied the role of dysferlin in Ca^2+^-induced G-actin incorporation into filaments in four different immortalized myoblast cell lines (DYSF2, DYSF3, AB320, and ER) derived from patients harboring mutations in the *dysferlin* gene. As compared with immortalized myoblasts obtained from a control subject, dysferlin expression and G-actin incorporation were significantly decreased in myoblasts from dysferlinopathy patients. Stable knockdown of dysferlin with specific shRNA in control myoblasts also significantly reduced G-actin incorporation. The impaired G-actin incorporation was restored by the expression of full-length dysferlin as well as dysferlin N-terminal or C-terminal regions, both of which contain three C2 domains. DYSF3 myoblasts also exhibited altered distribution of annexin A2, a dysferlin partner involved in actin remodeling. However, dysferlin N-terminal and C-terminal regions appeared to not fully restore such annexin A2 mislocation. Then, our results suggest that dysferlin regulates actin remodeling by a mechanism that does to not involve annexin A2.

## 1. Introduction

Dysferlin is a transmembrane protein containing seven cytosolic C2 domains, which bind Ca^2+^ and acidic phospholipids with different affinities [1,2]. Its most well-known function is to facilitate Ca^2+^-dependent aggregation and fusion of vesicles at wounded plasmalemma [3,4,5] by a mechanism that has not been completely elucidated. Dysferlin plays a role in other cellular processes in the skeletal muscle tissue including cytokine secretion and membrane receptor recycling in myoblasts [6,7] as well as biogenesis, remodeling, and maintenance of the T-tubule system [8,9]. Among the proteins involved in vesicle trafficking and/or membrane remodeling that interact with dysferlin are the SNAREs (acronym for soluble N-ethylmaleimide-sensitive factor activating protein receptors) syntaxin-4 and SNAP23 [10], annexins A1 and A2 [4], α-tubulin [11], and Mitsugumin 53 [12,13].

Mutations in the *dysferlin* gene cause a group of autosomal recessive muscular dystrophies known as dysferlinopathies [14]. The most common forms of dysferlinopathy are Miyoshi myopathy, limb-girdle muscular dystrophy type 2B, and distal anterior compartment myopathy [15,16]. Dysferlinopathy phenotypes include progressive atrophy of limb muscles, elevated serum creatine kinase levels, reduced expression of plasmalemmal dysferlin, and prevalence of immature muscle fibers [17,18]. As expected, skeletal muscle cells derived from dysferlinopathy patients [19] and dysferlin-deficient mice [3] display a defective Ca^2+^-dependent plasmalemma repair. Furthermore, when dysferlin expression is reduced, vesicles accumulate beneath the plasmalemma [20,21,22], suggesting a role of dysferlin in vesicle trafficking. A critical element for vesicle trafficking and membrane repair is the actin cytoskeleton [23,24,25,26,27]. Skeletal muscle cells express two cytoskeletal actin isoforms, β-actin and γ-actin, that localize in sub-plasmalemmal regions [28,29]. Skeletal muscle-specific ablation of β-actin or γ-actin causes a progressive myopathy, characterized by myofiber degeneration/regeneration and muscle weakness [29,30], thus emphasizing the critical role of the cytoskeletal actin network in the function of skeletal muscle cells.

Interplay between dysferlin and the actin cytoskeleton has been observed during plasmalemma repair [25,26]. Moreover, dysferlin interacts with proteins important for actin organization and remodeling such as annexin A2 [4], suggesting the implication of dysferlin in actin dynamics. Alterations in dysferlin expression, such as those occurring in dysferlinopathies, could then potentially affect actin dynamics in muscle cells. With this in mind, we studied whether the dynamics of the cytoskeletal actin is affected in myoblasts derived from skeletal muscle of dysferlinopathy patients. Our data show that the expression of dysferlin is dramatically reduced in dysferlinopathy-derived myoblasts compared to myoblasts from a healthy subject. Moreover, dysferlinopathy myoblasts exhibit a reduced capability to incorporate new actin monomers to the pre-existing actin filament (F-actin) network compared to control myoblasts, suggesting defects in actin cytoskeleton remodeling. Finally, the expression of a construct harboring the full-length dysferlin, as well as the expression of its N-terminal or its C-terminal regions, successfully restores actin dynamics in dysferlin-deficient myoblasts. These results support a role of dysferlin in actin cytoskeleton dynamics in muscle cells and suggest that this mechanism could be deregulated in dysferlinopathy.

## 2. Results

### 2.1. Dysferlin Expression in the Dysferlinopathy Cell Lines

Four different cell lines of immortalized myoblasts were derived from skeletal muscle biopsies from dysferlinopathy patients. These cell lines named DYSF2 (also called 107), DYSF3 (also called 379), AB320, and ER myoblasts were previously characterized [31,32]. Table 1 describes the origin of each cell line, including the mutations carried by donors. All of them are heterozygous with the exception of ER cells. As a control, we used the cell line C25, which was derived from a biopsy of semitendinosus muscle of a 25 year old male who did not suffer from any skeletal muscle disease [33]. All these cell lines were obtained from the platform for the immortalization of human cells from the Institut de Myologie (Paris, France), and their characterizations were previously reported [33,34,35]. All analyses were performed on undifferentiated myoblasts.

We first analyzed the relative expression of dysferlin in the cell lines DYSF2, DYSF3, AB320, and ER and compared it with the expression of the protein in C25 control myoblasts. Figure 1a shows a typical immunoblot stained with antibodies against dysferlin (upper bands) and β-tubulin (lower, loading control). Figure 1b shows dysferlin/β-tubulin ratios from five independent experiments. As compared with C25 myoblasts, the expression of dysferlin was reduced by 68%, 87%, 88%, and 83% in DYSF2, DYSF3, AB320, and ER myoblasts, respectively. These results agree with the expected dysferlin expression in these myoblasts [31,32,36,37] and validate these cell lines as in vitro models of dysferlinopathy.

### 2.2. Cytoskeleton Actin Dynamics in the Dysferlinopathy Myoblasts

The cytoskeletal actin network is a highly dynamic structure that is rearranged in skeletal muscle cells during vesicle trafficking and membrane repair [23,25,26,27,38,39], and such actin remodeling in skeletal myoblasts, evaluated as G-actin incorporation, seems to depend on high cytosolic Ca^2+^ concentrations [27]. Therefore, we measured the incorporation of actin monomers (G-actin) into pre-existing actin filaments in permeabilized myoblasts by using a previously reported assay [27,40]. In this assay, myoblasts were permeabilized with digitonin in a solution containing green-fluorescent Alexa-Fluor-488 G-actin and 2 mM ATP-Mg^2+^, in the absence or presence of Ca^2+^. As shown in the Appendix A, G-actin incorporation was significantly higher in the presence of 10 µM free Ca^2+^. Therefore, experiments were performed in this latter condition. Figure 2a shows examples of myoblasts with fluorescent actin filaments, indicative of incorporation of Alexa Fluor-labelled actin monomers into the F-actin network. Figure 2b shows the quantification of fluorescent G-actin incorporation. As compared with control C25 myoblasts, all dysferlinopathy myoblasts (DYSF2, DYSF3, AB320, and ER) displayed significant reduction in G-actin incorporation to the pre-existent cytoskeletal actin network (*p* < 0.05). In DYSF3 myoblasts, G-actin incorporation was reduced by 50%, whereas in the DYSF2, AB320, and ER myoblasts it was reduced by 36%, 35%, and 42%, respectively. Appendix A shows 1D intensity profiles of the incorporated fluorescently tagged G-actin. Higher-intensity fluorescence was observed in the cell periphery of C25 myoblasts, whereas an overall reduction of cell fluorescence was observed in the dysferlinopathy myoblasts (Appendix A). The peripheral distribution of the fluorescently-tagged G-actin probably corresponds to submembrane cortical cytoskeleton actin [28,29,40].

To rule out that the differences in G-actin incorporation in the control and dysferlinopathy myoblasts was a consequence of the size of pores formed by digitonin, we evaluated the incorporation of a protein with a size similar to G-actin into the cells. Therefore, we incubated digitonin-permeabilized C25 and DYSF3 myoblasts for 6 min at 37 °C in K^+^-glutamate /EGTA/Pipes (KGEP) buffer containing the GST-amphiphysin-1 SH3 domain fusion protein. Incorporation of GST-amphiphysin-1 SH3 was determined in fixed cells using a monoclonal anti-GST and secondary CY2-conjugated antibodies. As shown in Appendix A, no significant differences were encountered in the GST relative fluorescence intensity in C25 and DYSF3 cells.

We also performed experiments in RCMH myoblasts stably transfected with shRNA for dysferlin, a strategy to knockdown dysferlin expression previously reported by our group [32]. The RCMH cell line was established from a quadriceps biopsy of a healthy human [41], and it has been previously characterized [41,42]. The Appendix A shows the reduced expression of dysferlin in stably transfected RCMH myoblasts. As compared with non-transfected RCMH myoblasts, the stable knockdown of dysferlin significantly reduced G-actin incorporation into filaments (Figure 2c–d).

Together, these data strongly suggest that Ca^2+^-dependent G-actin incorporation is impaired when dysferlin is reduced, such as in the dysferlinopathy context.

### 2.3. Expression of Full-Length Dysferlin or Its C or N-Terminal Regions Restores G-Actin Incorporation in Dysferlinopathy Myoblasts

Next, we decided to evaluate whether the expression of dysferlin constructs can restore the impaired actin dynamics in DYSF3 myoblasts. We chose this myoblast cell line, and not the others, for its lower expression of dysferlin and reduced G-actin incorporation (Figure 1 and Figure 2). We first analyzed the incorporation of Alexa Fluor-labelled actin monomers in myoblasts expressing full-length dysferlin-HA. Representative images of these experiments are shown in Figure 3a. Quantification of fluorescent actin filaments in these cells shows that dysferlin-HA expression restored G-actin incorporation in DYSF3 myoblasts to levels comparable to C25 cells also expressing dysferlin-HA (Figure 3b).

We then evaluated how different dysferlin regions contributed to restore the impaired actin dynamics in the DYSF3 myoblasts. We specifically evaluated the N-terminal dysferlin region that comprises the Ca^2+^ and lipid-binding C2A, C2B, and C2C domains and FerI domain, and in parallel we examined the contribution of the C-terminal region containing a transmembrane region plus C2E, C2F, and C2G Ca^2+^-binding domains. Both dysferlin regions were fused to a mCherry reporter protein and transfected in C25 and DYSF3 myoblasts. Myoblasts expressing the empty mCherry vector were used as controls. Transfection efficiency was around 25%, 10%, and 35% in cells transfected with mCherry, dysferlin N-terminal, or C-terminal constructs. Fluorescence intensities of expression of these constructs are provided in the Appendix A.

A scheme of dysferlin domains is shown in Figure 4a. Images of fluorescent actin filaments and expression of the constructs are shown in Figure 4b. Figure 4c shows quantification of fluorescent actin filaments. The analyses show that either N-terminal or C-terminal parts of dysferlin increased actin fluorescence threefold in DYSF3 myoblasts, as compared to that observed in myoblasts expressing empty mCherry (Figure 4c). Furthermore, both dysferlin regions also increased G-actin incorporation by 1.5-fold to 1.8-fold in C25 myoblasts. This strongly suggests the involvement of both dysferlin regions in actin dynamics in skeletal myoblasts.

### 2.4. Annexin A2 Distribution in Dysferlin-Deficient Myoblasts

Annexin A2 is a Ca^2+^-binding protein that is recruited to membranes upon increments in cytosolic Ca^2+^, where it facilitates actin assembly [43,44]. Annexin A2 also associates to dysferlin in a Ca^+2^-dependent manner [4], and it is suggested that both are mutually required for their recruitment to injury sites [25,45]. Therefore, we analyzed the distribution of annexin A2 and its colocalization with newly incorporated G-actin in control C25 and dysferlinopathy DYSF3 myoblasts. These experiments were carried out in myoblasts permeabilized in the presence of 10 µM Ca^2+^ and Alexa Fluor-labelled G-actin, thus allowing the analysis of the colocalization of annexin A2 with the incorporated fluorescent G-actin. Figure 5a shows images of C25 and DYSF3 myoblasts immunostained with annexin A2 (red) and fluorescent actin (green). Annexin A2 localizes throughout the cytosol in C25 myoblasts, whereas it is mainly restricted to nuclear and perinuclear areas in DYSF3 myoblasts. Indeed, the ratio of the mean fluorescence intensity of the cytosol/whole cell of annexin A2 was 0.80 ± 0.02 in C25 myoblasts and 0.6 ± 0.03 in DYSF3 myoblasts (Figure 5b). Appendix A shows 1D intensity profiles of annexin A2 and fluorescently tagged G-actin. Both stains showed a similar distribution pattern in C25 myoblasts, whereas their distribution pattern was different in DYSF3 myoblasts, wherein annexin A2 showed a higher nuclear localization (Appendix A).

Pearson correlation coefficient analysis indicated a significant colocalization of annexin A2 with newly incorporated G-actin (0.65 ± 0.04; *n* = 35) in control C25 myoblasts, whereas this colocalization was lost in DYSF3 myoblasts (Pearson correlation coefficient of 0.34 ± 0.03; *n* = 32; Figure 5c).

### 2.5. Annexin A2 Distribution in Dysferlin-Deficient Myoblasts Expressing Dysferlin Constructs

Next, we determined whether the expression of full-length dysferlin-HA restores annexin A2 distribution in DYSF3 myoblasts. Figure 6A shows images of annexin A2 immunostaining (green) in C25 and DYSF3 myoblasts expressing full-length dysferlin-HA (red). Analyses of annexin A2 distribution are shown in Figure 6B. In DYSF3 myoblasts expressing full-length dysferlin-HA, annexin A2 cytosol/whole cell ratio was 0.76 ± 0.03 (*n* = 37), a value comparable with that of C25 myoblasts without transfection (0.80 ± 0.02). However, in C25 myoblasts expressing dysferlin-HA, annexin A2 cytosol/whole cell ratio was 0.46 ± 0.03. The change in annexin A2 distribution in the latter condition might be caused by the excess of dysferlin, since in mock condition this ratio was 0.86 ± 0.03 (Figure 6B). No colocalization of dysferlin with annexin A2 was observed in C25 or DYSF3 myoblasts expressing full-length dysferlin-HA (Pearson correlation coefficient of 0.4 ± 0.02 and 0.2 ± 0.03, respectively).

To determine whether dysferlin N- or C-terminal regions were capable of restoring annexin A2 mislocation in dysferlin-deficient myoblasts, we evaluated annexin A2 distribution in DYSF3 cells transfected with the mCherry constructs described above. Here, we included a control consisting of DYSF3 cells treated with the transfection reagents but without addition of the plasmid (mock). Figure 6C shows images of annexin A2 immunostaining (green) in DYSF3 myoblasts expressing the dysferlin N- or C-terminal constructs. Annexin A2 cytosol/whole cell ratios were 0.73 ± 0.02 (*n* = 45), 0.70 ± 0.04 (*n* = 22), and 0.77 ± 0.02 (*n* = 45) for DYSF3 myoblasts expressing mCherry or the N- or C-terminal constructs, respectively. For the mock experiment, the annexin A2 cytosol/whole cell ratio was 0.67 ± 0.04 (*n* = 25). Then, the annexin A2 cytosol/whole cell ratio in DYSF3 myoblasts expressing the C-terminal construct was comparable with that of C25 myoblasts without transfection (0.80 ± 0.02), and significantly different from the mock control, but not from the mCherry control (Figure 6d). No colocalization of annexin A2 with dysferlin N- or C-terminal was observed (Pearson correlation coefficient of 0.3 ± 0.03 and 0.2 ± 0.02, respectively).

Together, these data show that expression of full-length dysferlin restores annexin A2 distribution in dysferlinopathy myoblasts. However, the contribution of its C-terminal is not clear.

## 3. Discussion

Dysferlin and cytoskeletal actin play critical roles at various differentiation stages of skeletal muscle physiology; they are required for vesicle trafficking and exocytosis in myoblasts [6,7,27,46,47] and plasmalemma repair in myofibers [3,4,5]. Regarding the plasmalemma repair in the skeletal muscle, two mechanisms have been proposed. One of them includes Ca^+2^-induced recruitment of vesicles and other membranous organelles to repair sites, where they fuse together to form a membrane patch that seals the damage membrane [5,18]. The second mechanism involves Ca^+2^-induced exocytosis of lysosomes with the consequent release of acid sphingomyelinase, which in turn promotes the endocytosis of lesion areas [46,48]. It has been proposed that dysferlin is involved in both types of mechanisms, favoring vesicle recruitment and fusion in the first [5,18] and promoting lysosome exocytosis in the second [46]. The presence of dysferlin in vesicles or other membranous organelles seems to be important for such functions [5]. Annexin A2 is also critical for plasmalemma repair [4,49], and its ablation causes progressive muscle weakening [49]. Interplay between dysferlin, annexin A2, and actin remodeling has been observed during plasmalemmal repair. In this regard, actin filaments accumulate at membrane injury sites [26], and their disruption with cytochalasin D inhibits recruitment of dysferlin [25]. Likewise, Latrunculin A, a G-actin sequestering agent, delays the formation of the repair complex composed by annexins A1, A2, A5, and A6 [26]. In turn, annexin A2 facilitates actin remodeling [43] and contributes to the resealing of the plasma membrane by promoting actin polymerization [50]. Here, we found that dysferlin may also contribute to actin remodeling, as observed by a reduced G-actin incorporation into filaments in dysferlin-lacking myoblasts (Figure 2). Noteworthy, the poor G-actin incorporation in dysferlinopathy myoblasts was restored by expression of both N- and C-terminal regions of dysferlin (Figure 3); however, the contribution of these dysferlin regions to annexin A2 distribution remains unclear (Figure 6). Then, it is not possible to establish a causal relation between annexin A2 mislocation and impaired actin dynamics in dysferlin-deficient myoblasts.

Another critical element for actin remodeling is cytosolic Ca^2+^ level [51]. Indeed, high cytosolic Ca^2+^ concentrations promote both severing and formation of actin filaments [40]. The mechanism involves the participation of Ca^2+^-sensitive actin severing proteins, such as scinderin [52], as well as Src kinases and Rho GTPases, which lead to the activation of actin nucleation promoting factors such as N-WASP and cortactin [40,53,54]. Here, we observed that a high Ca^2+^ concentration (10 µM) favors G-actin incorporation in skeletal myoblasts (Appendix A), and such G-actin incorporation appears to be regulated by dysferlin Ca^+2^-sensitive C2 domains.

The fact that both N- and C-terminal regions of dysferlin restore actin dynamics suggests that dysferlin comprises functionally redundant modules. Both N- and C-terminal dysferlin constructs contain three C2 domains (Figure 4a), which bind Ca^2+^ and phospholipids with different affinity and Ca^2+^ dependency [2]. Synaptotagmin-1, another protein with tandem C2 domains, also promotes actin remodeling by binding phosphatidylinositol 4,5-bisphosphate (PI(4,5)P_2_) via a polybasic region [55]. Reportedly, PI(4,5)P_2_ promotes actin remodeling by recruiting actin binding proteins, such as formins [56,57], a family of proteins involved in actin nucleation and assembly [58]. Dysferlin also binds and recruits PI(4,5)P_2_ to membranes and promotes membrane remodeling and T-tubule biogenesis in a PI(4,5)P_2_-dependent manner [59]. Then, the interaction of dysferlin C2 domains with PI(4,5)P_2_ might explain its ability to remodel the actin cytoskeleton. However, only the isolated C2A domain of dysferlin binds PI(4,5)P_2_ in vitro in a Ca^2+^-dependent way, whereas C2F binds weakly to phosphatidylinositols. The other isolated C2 domains exhibited no detectable associations [1]. Since the study of Therrien et al. (2009) [1] was carried out with isolated C2 domains, it did not consider potential cooperative effects between tandem C2 domains, as reported for synaptotagmin-1 C2 domains for its association to PI(4,5)P_2_ [60]. Then, further studies are necessary to achieve a better comprehension of the mechanism by which dysferlin promotes actin dynamics.

Full-length dysferlin-HA, as well as the dysferlin N-terminal and C-terminal constructs, exhibit a granular cytoplasmic pattern. This granular pattern might reflect the distribution pattern of dysferlin, but it might also be determined by the distribution pattern of the constructs, since mCherry seems to form aggregates (Figure 4B). A granular cytoplasmic expression of dysferlin has been observed in myoblasts [46], differentiated L6 myotubes [5], and skeletal myofibers [61]. PI(4,5)P_2_ also associates to intracellular compartments, such as vesicles, lysosomes, endosomes, and the nucleus, among others [62,63]. Then, dysferlin might favor PI(4,5)P_2_-dependent actin remodeling in intracellular compartments, contributing to the described dysferlin-dependent processes in myoblasts, such as vesicle trafficking and exocytosis [7,46]. However, additional studies are necessary to demonstrate the role of dysferlin in actin remodeling during plasmalemmal repair in differentiated skeletal muscle cells, since our study was performed in digitonin-permeabilized myoblasts, and dysferlin seems to be unable to repair membrane damage induced by nonionic detergents [64].

This work might contribute to a better understanding of the mechanisms involved in the pathogenesis of muscular dystrophies caused by mutations in dysferlin.

## 4. Materials and Methods

### 4.1. Reagents

Actin, from rabbit muscle, Alexa Fluor™ 488 conjugate (Thermo Fisher Scientific, Waltham, MA, USA); ATP (Sigma-Aldrich, St. Louis, MO, USA); bovine serum albumin (Sigma-Aldrich, St. Louis, MO, USA); dexamethasone (Sigma-Aldrich, St. Louis, MO, USA); 40,6-diamidino-2-phenylindole (Sigma-Aldrich, St. Louis, MO, USA); digitonin (Sigma-Aldrich St. Louis, MO, USA); dimethyl sulfoxide (Merck Company, Darmstadt, Germany); DMEM/F-12 medium (Gibco, BRL, Gaithersburg, MD, USA); Dulbecco modified Eagle’s minimal essential medium (Gibco BRL, Gaithersburg, MD, USA); EDTA (Calbiochem, La Jolla, CA); EGTA (Sigma-Aldrich, St. Louis, MO, USA); fetal bovine serum (Gibco BRL, Gaithersburg, MD, USA); fetuin (Sigma-Aldrich, St. Louis, MO, USA); gentamicin (Gibco/Life Technology, China); glutamic acid (Sigma-Aldrich, St. Louis, MO, USA); HEPES (Calbiochem, La Jolla, CA); human insulin (Eli Lilly and company, Indianapolis, USA); Lipofectamine 2000 (Invitrogen, Carlsbad, CA, USA); 199 medium (Sigma-Aldrich St. Louis, MO, USA); NaF (Merck Company, Darmstadt, Germany); Na_3_VO_4_ (Merck Company, Darmstadt, Germany); Opti-Mem (Gibco BRL, Gaithersburg, MD, USA); penicillin (OPKO, Santiago, Chile); p-formaldehyde (Sigma-Aldrich St. Louis, MO, USA); phenylmethyl sulfonylflouride (Sigma-Aldrich, St. Louis, MO, USA); PIPES (Sigma-Aldrich, St. Louis, MO, USA); poly-l-lysine (Sigma-Aldrich, St. Louis, MO, USA); recombinant human basic fibroblast growth factor (Gibco BRL, Gaithersburg, MD, USA); protease inhibitor cocktail (Sigma-Aldrich, St. Louis, MO, USA); recombinant human epidermal growth factor (Gibco BRL, Gaithersburg, MD, USA); triton X-100 (Merck Company, Darmstadt, Germany); trypsin- EDTA 0.25% (Sigma-Aldrich, St. Louis, MO, USA); and Tween-20 (Merck Company, Darmstadt, Germany) were purchased.

### 4.2. Antibodies

Anti-annexin A2 (Santa Cruz Biotechnology, Inc, Dallas, Texas, USA); anti-β-tubulin (Cytoskeleton, St. Denver, CO, USA); anti-dysferlin-HAMLET (Novocastra TM Lyophilized Leica; Newcastle, United Kingdom); CY2-conjugated goat anti-rabbit IgG (H+L) (Jackson Immunoresearch, West Grove, PA, USA); Cy3-conjugated goat anti-rabbit IgG (H+L) (Jackson Immunoresearch, West Grove, PA, USA); horseradish peroxidase (HRP)-conjugated donkey anti-sheep (R & D Systems, Minneapolis, USA); HRP-conjugated goat anti-mouse IgG (H+L) (Thermo Fisher Scientific, Waltham, MA, USA); and peroxidase affiniPure F(ab’)_2_ fragment donkey anti-rabbit IgG (H+L) (Jackson ImmunoResearch, West Grove, PA, USA) were purchased.

### 4.3. cDNA Constructs and Plasmids

Plasmid dysferlin-HA (Addgene plasmid 29767) was provided by The Jain Foundation (www.jain-foundation.org). The mCherry-tagged plasmids containing dysferlin N-terminal (amino acids 1–572) or C-terminal (amino acids 1169–2119) were constructed by GenScript Corporation (Nanjing, China) by cloning the appropriate DNA sequences (Homo sapiens dysferlin isoform 1, NCBI; reference Sequence: NP_001124459.1) into a mCherry_pcDNA3.1(+) vector. mCherry was fused to the C-terminal end in dysferlin N-terminal and fused to the N-terminal end in dysferlin C-terminal. The plasmid encoding for the amphiphysin SH3 domain fused to a glutathione-S-transferase (GST)-tag (amphiphysin-SH3) was provided by Dr. Patricia Hidalgo (Institut für Neurophysiologie, Medizinische Hochschule Hannover, Germany). shRNA plasmids against dysferlin with puromycin resistance were obtained from Santa Cruz Biotech (Dallas, Tx).

### 4.4. Culture of Cell Lines and Transfection

The C25 cell line was established from human biopsies of semitendinosus of unaffected individuals [33,35]. The dysferlinopathy cell lines DYSF2, DYSF3, AB320, and ER were established from human biopsies of patients bearing different dysferlin mutations (c.855  +  1delG c.895G  >  A; c.1448C  >  A c.107 T  >  A; c.342-1G  >  A HTZ c.3516_3517delTT (p.Ser1173X) HTZ; G1628R (c.4882G  >  A) HMZ, respectively). All these cell lines (C25, DYSF2, DYSF3, AB320, and ER myoblasts) were obtained from the platform for the immortalization of human cells from the Institut de Myologie (Paris, France), with agreement of the subjects through signature of an informed consent and anonymization before immortalization, according to the EU GDPR regulation. Their characterization has been previously reported [31,33,35]. They were cultivated in a mix of 199 medium/Dulbecco modified Eagles minimal essential medium (1:4 ratio) supplemented with 20% fetal bovine serum (FBS), 25 μg/mL fetuin, 0.5 ng/mL basic fibroblast growth factor, 5 ng/mL epidermal growth factor, 0.2 μg/mL dexamethasone, 5 μg/mL insulin, 50 U/mL penicillin, and 100 µg/mL gentamicin at a density of 3 × 105 cells/mL in 25 mm glass coverslips and incubated at 37 °C in a 5% CO_2_ atmosphere until experimentation. For transfections, cells were incubated in 50 µL of Opti-Mem containing 1 µg DNA and 1.5 µL of Lipofectamine 2000 for 20 min. Then, 250 µL of Opti-Mem was added, and cells were kept at 37 °C in a 5% CO_2_ atmosphere for 5 h. After that period, 1 mL of the culture medium described above was added, and cells were kept at 37 °C in a 5% CO_2_ atmosphere for 24 h. 

RCMH myoblasts were cultured in DMEM/F-12 medium supplemented with 10% fetal bovine serum and incubated at 37 °C in a 5% CO_2_ atmosphere. Transfection of shRNA plasmid against dysferlin with puromycin resistance was performed with Lipofectamine 2000 as described above. A stable transfected pool was selected with 10 µg/mL puromycin in the cultured medium.

### 4.5. Immunoblotting Analyses

Cells were lysed in a non-denaturing lysis buffer composed of 300 mM NaCl, 5 mM EDTA, 50 mM TRIS HCl, 1% Triton X-100, and supplemented with 0.1% (*v*/*v*) protease inhibitor cocktail, 50 mM NaF, and 0.2 mM Na_3_VO_4_. Total protein content was determined using the Quant-it Protein Assay Kit (Invitrogen, Carlsbad, CA, USA). Total proteins (100 μg) were separated by sodium dodecyl sulfate-polyacrylamide gel electrophoresis (10% polyacrylamide gels) and electrophoretically transferred to polyvinylidene difluoride membranes (GE Healthcare Life Sciences, Piscataway, NJ, USA). Blots were first incubated with Tris-buffered saline containing 5% bovine serum albumin and 1% Tween-20 for 1 h at room temperature. Then, the membranes were cut at approximately 40 kDa for parallel incubation with the antibody against β-tubulin (control loading) or the tested antibody. Afterwards, membranes were incubated overnight at 4 °C with a specific antibody against dysferlin (1:500), or β-tubulin (1:1000). Later, membranes were washed and incubated with a secondary antibody (anti-rabbit HRP, 1:5000 or anti-sheep HRP, 1:2500) for 1 h. Immunoreactive bands were detected using ECL Select Western Blotting Detection Reagent (GE Healthcare Bio-Sciences Corp., Piscataway, NJ, USA) and an image acquisition system Epichemi3 Darkroom (Cambridge Scientific, Watertown, MA, USA). ImageJ 1.43 m (NIH, Bethesda, MD, USA) was used for quantification. To measure the relative expression levels of dysferlin in the cell lines, the signal intensity of dysferlin bands obtained after background subtractions were normalized with respect to the signal intensity of the loading control (β-tubulin) detected on the same blot.

### 4.6. G-actin Incorporation into Filaments and Immunofluorescence

Non-transfected or transfected myoblasts were incubated 6 min at 37 °C in KGEP buffer (139 mM K^+^ glutamate, 20 mM PIPES, 5 mM EGTA, 2 mM ATP-Mg^2+^, 0.3 μM Alexa Fluor 488-G-actin conjugate, and 20 μM digitonin, pH 6.9) in the absence or presence of 10 µM free Ca^2+^ [27,40]. The online software Ca-EGTA Calculator v1.2 (University of California, Davis, CA, USA; https://somapp.ucdmc.ucdavis.edu/pharmacology/bers/maxchelator/CaEGTA-NIST.htm) was used to estimate the Ca^+2^ and EGTA concentrations to achieve 10 µM free Ca^2+^ with parameters of pH of 6.9 and temperature equal to 25 °C. Next, samples were fixed with 4% p-formaldehyde (PFA), stained with 5 mg/mL 4,6-diamidino-2-phenylindole (DAPI), and visualized by confocal microscopy. In cells transfected with dysferlin-HA, after PFA fixation, samples were incubated with the anti-dysferlin antibody (1:50), washed three times, and incubated with a Cy3-conjugated anti-rabbit secondary antibody (1:100). For immunodetection of annexin A2, after PFA fixation, samples were incubated with the anti-annexin A2 antibody, washed three times, and incubated with Cy2-conjugated anti-rabbit secondary antibody (1: 250). Images were captured at the equatorial plane of the cells using identical exposure settings between compared samples. To quantify G-actin incorporation, cell fluorescence of each individual cell was divided by the cell area, which was determined by manually drawing the cell outline using the differential interference contrast. For annexin A2 distribution, annexin A2 immunostaining was analyzed also in a single focal plane, and data are shown as the ratio of the mean fluorescence intensity of the cytosol/whole cell of annexin A2. All images were analyzed and processed using ImageJ software (NIH, Bethesda, USA).

### 4.7. Statistics

Results were expressed as means ± SEM. Normality of data was checked using the Kolmogorov–Smirnov test. Statistical comparisons were performed using a *t*-test. All statistical analyses were performed using InStat3 software (GraphPad Software Inc, La Jolla, CA, USA).

### 4.8. Ethics Statement

The investigators declare to know the Manual of Biosafety Regulations stipulated by CONICYT (Chile), version 2008; CDC (USA) Biosafety Manual 4th Edition; and Laboratory Biosafety, WHO, Geneva, 2005 (mainly in reference to experiments with recombinant DNA and RNA and the manipulation of cell lines). This research was approved by the Biosafety and Bioethics committees of Universidad de Valparaíso (Chile), approval identification numbers BS002/2016 and BEA080-216, respectively.

## Figures and Tables

**Figure 1 ijms-21-00037-f001:**
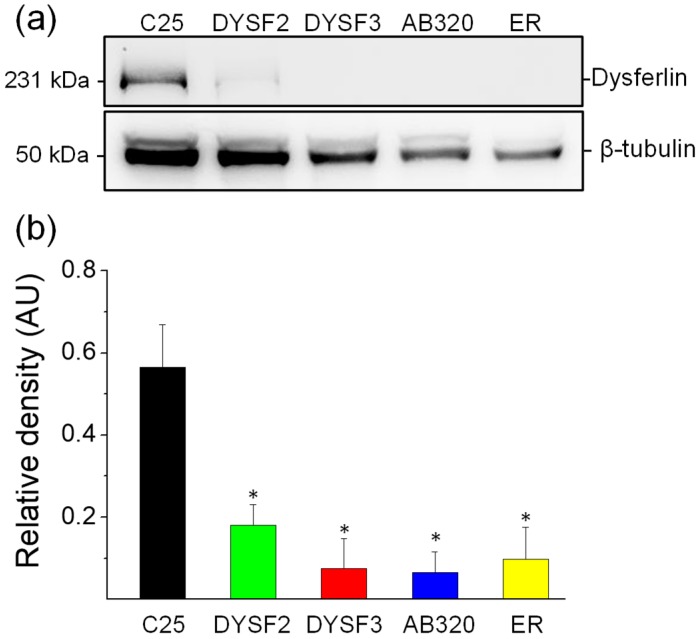
Reduced expression of dysferlin in immortalized myoblasts obtained from dysferlinopathy patients. Dysferlin expression was determined by immunoblotting of total protein extracts from C25 myoblasts obtained from an unaffected donor, or DYSF2, DYSF3, AB320, or ER myoblasts obtained from dysferlinopathy patients. (**a**) Example of immunoblot detection of dysferlin (upper bands) and β-tubulin (loading control; bottom bands). (**b**) Relative density (dysferlin/β-tubulin ratio). Data are means ± SEM from 5 independent immunoblots. * *p* < 0.05 compared to C25 myoblasts (*t*-test).

**Figure 2 ijms-21-00037-f002:**
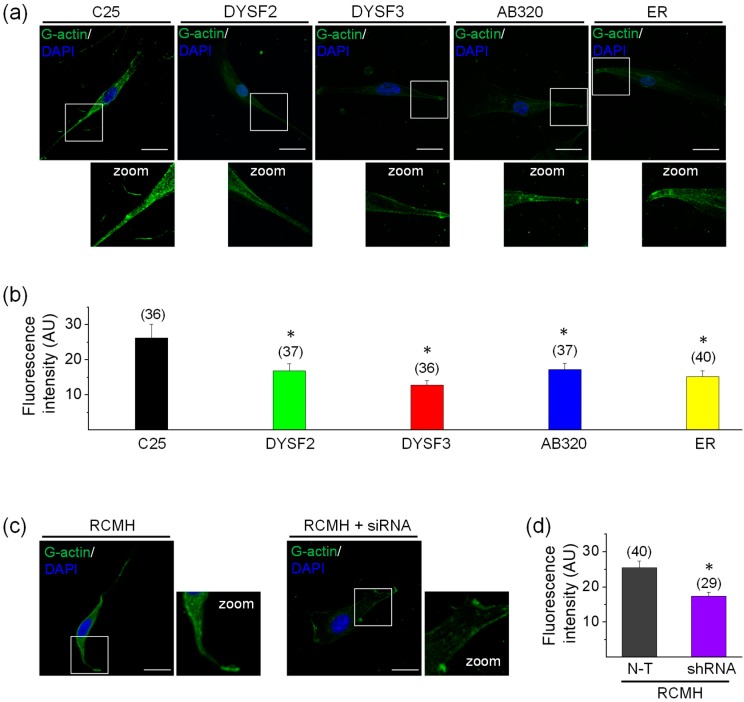
G-actin incorporation into filaments in dysferlin-deficient myoblasts. Fluorescently tagged G-actin incorporation into filaments was assayed in myoblasts permeabilized with 20 µM digitonin in the presence of 300 nM Alexa-Fluor-488 actin, 2 mM ATP-Mg^2+^, and 10 µM free Ca^2+^ during 6 min at 37 °C. After permeabilization, cells were fixed and nuclei were stained with DAPI. Confocal images were acquired at the equatorial plane of the cells using identical exposure settings between compared samples. (**a**–**c**) Fluorescent actin filaments in control C25 and dysferlinopathy DYSF2, DYSF3, AB320, and ER myoblasts (**a**) and RCMH myoblasts non-transfected (N-T) or stably transfected with shRNA for dysferlin (**c**). Scale bar = 20 µm. Insets show digital zooms of the boxed areas; brightness and contrast were increased to appreciate better the pattern of fluorescent actin. (**b**–**d**) Data are means ± SEM. Actin fluorescence intensity was measured in a single focal plane at the equator of cells and normalized by the cell area. The number of analyzed cells from five different cultures is indicated in parentheses. * *p* < 0.05 compared to C25 (b) or N-T RCMH myoblasts (c) (*t*-test).

**Figure 3 ijms-21-00037-f003:**
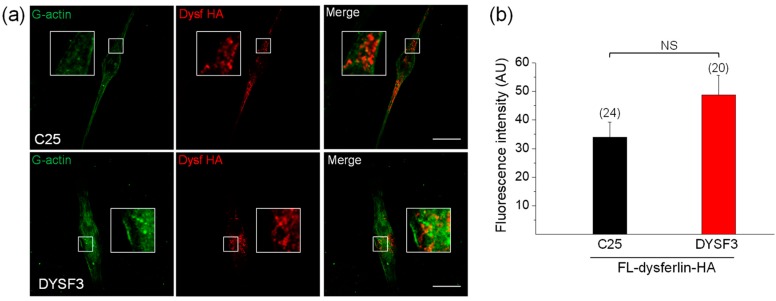
Full-length dysferlin-HA restores G-actin incorporation in dysferlinopathy myoblasts. Control C25 or dysferlinopathy DYSF3 myoblasts were transfected with full-length dysferlin-HA (FL-dysferlin-HA), and 24 h fluorescently tagged G-actin incorporation was assayed as previously described. FL-dysferlin-HA expression was assayed by immunofluorescence using a monoclonal antibody against dysferlin and a Cy3-conjugated anti-rabbit secondary antibody. Confocal images were acquired at the equatorial plane of the cells using identical exposure settings between compared samples. (**a**) C25 and DYSF3 myoblasts with fluorescent actin filaments (green) and FL-dysferlin-HA immunostaining (red). Scale bar = 10 µm. Insets show digital magnification of the boxed areas. (**b**) Bars represent means ± SEM. Actin fluorescence intensity was measured in a single focal plane at the equator of cells and normalized by the cell area. The number of cells analyzed from four different cultures is indicated in parentheses. No significant differences (NS) were found (*t*-test).

**Figure 4 ijms-21-00037-f004:**
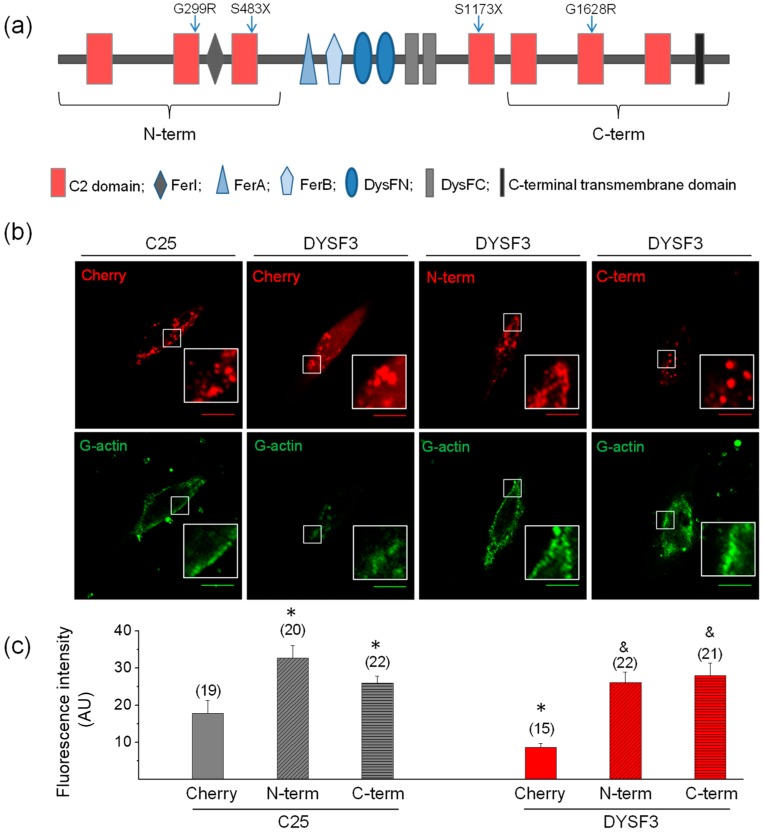
The N- or C- terminal dysferlin regions are efficient in restoring G-actin incorporation in dysferlinopathy myoblasts. (**a**) Schematic representation of dysferlin regions included in the N-terminal and C-terminal constructs. Blue arrows indicate positions of dysferlin mutations carried by the cell lines (see Table 1). (**b**,**c**) Control C25 or dysferlinopathy DYSF3 myoblasts were transfected with mCherry alone (Cherry), or dysferlin N-terminal (N-term) or C-terminal (C-term) fused to mCherry. Twenty-four hours later, fluorescently tagged G-actin incorporation was assayed as described in Methods, and confocal images were acquired at the equatorial plane of the cells using identical exposure settings between compared samples. (**b**) Fluorescent actin filaments in C25 or DYSF3 myoblasts expressing Cherry, and DYSF3 myoblasts expressing N-term or C-term. Scale bar = 10 µm. Insets show digital magnification of the boxed areas. (**c**) Actin fluorescence intensity was measured in a single focal plane and normalized by the cell area. Bars represent means ± SEM. Actin fluorescence intensity was measured in a single focal plane at the equator of cells and normalized by the cell area. The number of analyzed cells from four different cultures is indicated in parentheses. * *p* < 0.05 compared with C25 myoblasts expressing mCherry, ^&^
*p* < 0.05 compared with DYSF3 expressing mCherry (*t*-test).

**Figure 5 ijms-21-00037-f005:**
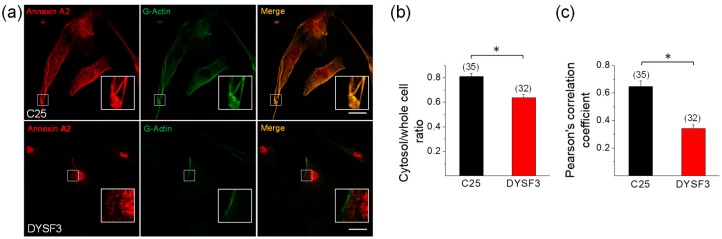
Annexin A2 distribution in dysferlinopathy myoblasts. Distribution of annexin A2 was analyzed by immunofluorescence using an anti-annexin A2 antibody and Cy3-conjugated anti-rabbit secondary antibody in C25 and DYSF3 myoblasts permeabilized with 20 µM digitonin in the presence of 300 nM Alexa-Fluor-488 actin, 2 mM ATP-Mg^2+^, and 10 µM free Ca^2+^. Confocal images were captured at the equatorial plane of the cells using identical exposure settings between compared samples; therefore, annexin A2 distribution was measured in a single focal plane. (**a**) C25 and DYSF3 myoblasts immunostained with annexin A2 (red) and fluorescent G-actin incorporated into filaments (green). Scale bar = 20 µm. Insets show digital magnification of the boxed areas. (**b**,**c**) Bars represent means ± SEM of the ratio of the mean fluorescence intensity of the cytosol/whole cell of annexin A2 immunostaining (**b**) and Pearson correlation coefficient for colocalization of annexin A2 with fluorescent actin filaments (**c**). The number of analyzed cells from four different cultures is indicated in parentheses. * *p* < 0.05 (*t*-test).

**Figure 6 ijms-21-00037-f006:**
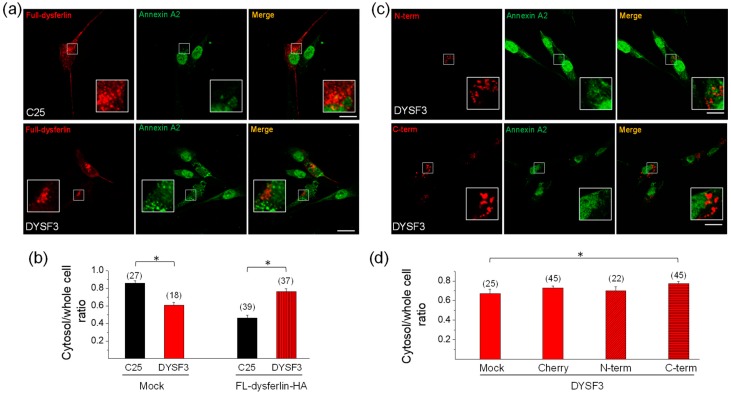
Annexin A2 distribution in DYSF3 myoblasts expressing dysferlin constructs. Distribution of annexin A2 was analyzed by immunofluorescence using an anti-annexin A2 antibody and Cy2-conjugated anti-rabbit secondary antibody in C25 and DYSF3 myoblasts expressing full-length dysferlin-HA (FL-dysferlin-HA) (**a**,**b**) and mCherry (Cherry), or dysferlin N-terminal (N-term) or C-terminal (C-term) fused to mCherry (**c**,**d**). Experiments were performed in digitonin-permeabilized cells in the presence of 10 µM free Ca^2+^. Confocal images were captured at the equatorial plane of the cells using identical exposure settings between compared samples; therefore, annexin A2 distribution was measured in a single focal plane. (**a**,**c**) Annexin A2 stained (green) in C25 or DYSF3 myoblasts in a mock condition or expressing FL-dysferlin-HA (**a**) and N-term or C-term (**c**). Scale bar = 20 µm. Insets show digital magnification of the boxed areas. (**b**,**d**) Data show means ± SEM of the ratio of the mean fluorescence intensity of the cytosol/whole cell of annexin A2 immunostaining. The number of analyzed cells from four different cultures is indicated in parentheses. * *p* < 0.05 (*t*-test).

**Table 1 ijms-21-00037-t001:** Description of the immortalized human skeletal myoblasts used in this study.

Cell Line	Muscle Biopsy	Patient	Mutations
DYSF2	Vastus lateralis	37 year old male	Exon 8:c.855 + 1delG, mRNAdecay Exon 9: c.895G > A, r.895G > A, p.G299R
DYSF3	Vastus lateralis	36 year old female	Exon 16: c.1448C > A, p.S483X Exon 55:c.*107T > A, 3′UTR
AB320	Quadriceps	29 year old female	Intron 4: c.342-1G > A Exon 32: c.3516–3517delTT, p.S1173X
ER	Quadriceps	17 year old male	Homozygous Exon 44: c.4882G > A, p.G1628R

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
