# Peer review of "Defects in G-Actin Incorporation into Filaments in Myoblasts Derived from Dysferlinopathy Patients Are Restored by Dysferlin C2 Domains"

_ijms, 2019, doi:10.3390/ijms21010037_

Round 1

Reviewer 1 Report

The manuscript by Ximena Baez-Matus and co-authors reports that myoblasts derived from patients with dysferlinopathy demonstrate a reduced level of monomeric actin incorporation in the actin cytoskeleton as compared to normal myoblasts. The authors also demonstrate that actin dynamics, but not annexin A2 localization, can be rescued by either N- or C-terminal dysferlin halves.

The manuscript addresses a potentially novel aspect in the pathogenesis of dysferlinopathy, suggesting that the reduction in the dysferlin content may reduce the Ca2+-dependent actin cytoskeleton reorganization and thus lead to deficiencies in myoblast regeneration. Although the work is in interesting and potentially important, it is missing important controls.

Major comments:

To confirm the role of Ca2+ in the observed actin reorganization, the experiments should be conducted in the absence of Ca2+ and the presence of a Ca2+ chelator (EGTA) in the medium. Formation of pores by digitonin depends on the cholesterol level in the cytoplasmic membrane. Given that dysferlin is a membrane-associated protein involved in membrane homeostasis and that the lipid composition (including the cholesterol content) of skeletal muscle is altered in dysferlinopathy (PMID: PMC6672035), it is conceivable that the pores formed in normal and affected myoblasts by digitonin may vary in size and stability. Such scenario is perhaps even more likely considering that dysferlin is an interacting partner of caveolin, whose role in cholesterol homeostasis regulation is also recognized (PMID: 16603689). Differences in pore size and stability may, in turn, cause differences in Ca2+ and actin entry to the cell and may account for the differences in incorporation of the labeled actin into the actin network. To account for such possibility, the authors should check whether cell-impermeable molecules of ~ actin size (e.g., fluorescent dextrans) show identical cell entry kinetics to control and dysferlin-deficient myoblasts. It is not clear how the fluorescence levels of the digitonin-treated cells were measured. Has the fluorescence of the entire cell been integrated? If the cell size and shape differ substantially between dysferlin-deficient and normal myoblasts, such calculations may lead to errors and must be taken thoughtfully and described clearly. Does transfection with full-length dysferlin rescue the localization of Annexin 2? This control is missing. Images of the cells should be of higher resolution and larger scale. Figure 1a: the samples should be better normalized – the tubulin content differs at least by five-fold between C25 and ER myoblasts. Even though the ratiometric relative density was implemented, low levels of dysferlin may fall from the linear range of the detection method and therefore could report false values.

Minor comments:

The selection of buffer conditions appears to be poorly optimized for the goals of the experiment. Particularly, 5mM EGTA is unreasonably high to buffer Ca2+ to 10uM final free concentration. Under these conditions, pH change by only 0.2 units (from 6.9 to 7.1) would reduce [Ca2+] two-fold and a small (~10%) inaccuracy in total [Ca2+] would drop the final free Ca2+ levels by nearly four-fold. Reducing EGTA to submillimolar levels could be recommended. It is not indicated how the concentration of free Ca2+ was calculated. Figure 2 and 3 legends: “G-actina” instead of “G-actin” Figure 3b legend: “FL-dyferlin”; "s" is missing. A brief discussion of recognized mechanisms of dysferlinopathy (e.g., the role of lysosome-cytolemma fusion and ASM release) would be appropriate and desirable. Similarly, discussion of the mechanisms behind the Ca2+ effects on the actin cytoskeleton reorganization would be desirable. Repair of the Saponin-caused muscle damage is not affected by dysferlin (PMID 21941430). Since digitonin causes similar membrane damage, how relevant is this report to the current study?

Reviewer 2 Report

The authors show that cells lacking dysferlin incorporate less G-actin into pre-existing actin filaments in permeabilized myoblasts, that either the C or N terminal domains of dysferlin can confer this activity, and that annexin 2 does not seem to be relevant. The results may be of interest to others studying this protein’s role in dysferlinopathies .

Major concerns:

Patient-derived cell lines may have many differences from each other. It would be cleaner to compare control and dysferlin deleted cell lines (perhaps generated with CRISPR).

Actin ‘dynamics’ are not measured in the paper. The paper would be improved by the measurement of actin dynamics using a live-label such as LifeAct, perhaps in combination with FRAP. At the least, the wording in the title and abstract should be changed to more accurately reflect the experiments performed.

Clarity of the figures should be improved. Specifically:

Fig. 2. While actin incorporation does appear to be reduced, the images are not at sufficient magnification to show discernable actin structures. Please provide larger (zoomed in) images at higher magnification. 2B A correction for multiple comparisons should be applied.

Fig. 3. While actin incorporation does appear to be rescued, the images are not at sufficient magnification to show discernable actin structures. Please provide larger (zoomed in) images at higher magnification. I also don’t understand why the Ns are so small.

Fig. 4. I can’t see anything in the images. Please zoom in and brighten the images so structures can be discerned. Why is dysferin in the nucleus in the top left panel of 4D but not the other panels in D? Why is annexin showing up in the nucleus in D but not in A?

I am confused by the rescue by both the N- and C- terminal regions of dysferlin. The authors should explain why plasma membrane localization might not be important for rescue.

The paper would benefit from a more thorough analysis of the mechanistic connection between dysferlin and actin polymerization. For example, perhaps a formin is involved?  

Minor:

The writing is wordy throughout. I recommend thorough editing for conciseness. A few examples:

Line 20

 ‘the dynamics of the cytoskeletal actin’ shoud be actin dynamics

Line 40 ‘Dysferlin is also proposed to be required for other’ should be dysferlin plays a role in other

Line 54 ‘Furthermore, ultrastructural analysis have evidenced accumulation of vesicles beneath the plasmalemma of cells with reduced dysferlin expression’ should be When dysferlin expression is reduced, vesicles accumulate beneath the plasmalemma.

Minor typos

62 dysferlin and the actin cytoskeleton

108 The ‘cytoskeleton’ network à the cytoskeletal network

Reviewer 3 Report

This study investigated the hypothesis that dysferlin regulates actin dynamics in myoblasts.  Specifically, the authors used dysferlinopathy patient-derived myoblasts, which had reduced dysferlin expression, to test if Ca2+-dependent actin polymerization was affected in these cells.  They further determined the domain in dysferlin required for this function.  They concluded that regulation of actin by dysferlin involves both N- and C-terminal regions of dysferlin but is independent of Annexin A2.  Dissecting the structural requirement for regulation of actin dynamics is important and of interests to several fields given the links to phosphoinositides, membrane trafficking and cell motility.  However, there are a few major issues in the study that should be addressed before acceptance.

Major issues:

How was the amount of G-actin incorporation measured? Was it measured throughout the whole volume of cells or a single focal plane?  The authors need to describe in details how the image analysis was done.  As actin structures vary widely in different locations in the cell, it is critical to know whether dysferlin affects actin polymerization globally in the whole cell or a specific subcellular localization.  If only one focal plane was chosen for the analysis, authors need to provide a way validating it is the same/similar focal plane for control and the reduced dysferlin cells.  

Along the same rationale to point 1., details about how the mislocalization of Annexin A2 was quantified (cytosol/whole cell) needs to be described.

In Fig. 3, what are the expression levels of full-length, N- and C-termini of dysferlin? The difference in their expression may change the conclusion of either N- or C-terminal regions is sufficient for dysferlin’s function.

In Fig. 4, the difference in Annexin A2 localization between C25 and DYSF3 cells with Cherry expression should be done side by side with the N- and C-terminal fragments of dysferlin. As in 4B, C25 ratio of 0.8 is very close to the DYSF3+Cherry ratio of 0.7 in 4E.  What is the ratio in C25+Cherry? 

Reduced dysferlin had a significant effect on the colocalization between Annexin A2 and G-actin. How was this (Person’s correlation coefficient) affected by the full-length, N- and C-termini of dysferlin?  This is critical in assessing whether dysferlin regulation of actin depends on Annexin A2. 

Minor issue:

It would be informative to annotate the positions of mutations of each patient-derived cell lines on the domain structure of dysferlin in Fig. 3.

Round 2

Reviewer 1 Report

The authors of the manuscript have adequately addressed most of my concerns, except a minor issue with misspelling "G-actina" in Figs 3 and 4.

The updated higher magnification images provide a better view of the reported effects. The comprehension of the information could be further improved by providing 1D intensity plots in all the fluorescence channels at the representative sections of the cells. Such plots, especially if more than one is provided for each experimental condition, would allow demonstrating clearly whether the protein distribution is restricted to particular areas (membrane, nucleus, cytoplasmic granules) or spread through the entire cytoplasm.

It would also be desirable to check the localization of F-actin in the analyzed cells. It is possible that the observed difference in the incorporation of G-actin is dictated by a difference in the localization of F-actin. Although such experiments would be very desirable, I would leave the final decision to the discretion of the authors.

Reviewer 2 Report

The authors have submitted a much improved version of the paper, and have addressed my concerns. The insets (zooms) in the images are very helpful - it is now possible for the reader to see the actin structures being discussed. 

Two minor comments:

mCherry is notorious for forming aggregates, and indeed, the mCherry images do appear quite blobby. The authors may want to comment on whether the mCherry tag itself is contributing to the 'granular' appearance they observe. 

Fig. 3,4 still contain typos (G-actina). 

Reviewer 3 Report

The revised manuscript is significantly improved and now includes information of dysferlin knockdown effect on G-actin incorporation and descriptions on the methods, among others.  One remaining issue is the rescue experiments on annexin A2 distribution.  One critical comparison should be between C25 cells+mock transfection and DYSF3+mock.  If these two groups are not different from each other, there is no phenotype to rescue to begin with.  In addition, mCherry alone had a similar ratio to C-terminus of dysferlin, so it seemed that every construct used in the experiments “rescue.”  Together, the conclusion of full-length dysferlin restores annexin A2 distribution needs to be reconsidered based on the comparison between C25 cells+mock transfection and DYSF3+mock.

There is a typo: Figure S3 should be Figure S2.
